# Brain Tumor Analysis Using Deep Learning and VGG-16 Ensembling Learning Approaches

**Ayesha Younis** [1] , **Li Qiang** [1,*] **, Charles Okanda Nyatega** [2,3] **, Mohammed Jajere Adamu** [1]
**and Halima Bello Kawuwa** [4]

1   School of Microelectronics, Tianjin University, Tianjin 300072, China; ayesha@tju.edu.cn (A.Y.);
    mainajajere@tju.edu.cn (M.J.A.)
2   School of Electrical and Information Engineering, Tianjin University, Tianjin 300072, China;
    ncharlz@tju.edu.cn
3   Department of Electronics and Telecommunication Engineering, Mbeya University of Science and Technology,
    Mbeya P.O. Box 131, Tanzania
4   School of Precision Instrument and Opto-Electronics Engineering, Tianjin University, Tianjin 300072, China;
    halima@tju.edu.cn
*   Correspondence: liqiang@tju.edu.cn

**Abstract:** A brain tumor is a distorted tissue wherein cells replicate rapidly and indefinitely, with no control over tumor growth. Deep learning has been argued to have the potential to overcome the challenges associated with detecting and intervening in brain tumors. It is well established that the segmentation method can be used to remove abnormal tumor regions from the brain, as this is one of the advanced technological classification and detection tools. In the case of brain tumors, early disease detection can be achieved effectively using reliable advanced A.I. and Neural Network classification algorithms. This study aimed to critically analyze the proposed literature solutions, use the Visual Geometry Group (VGG 16) for discovering brain tumors, implement a convolutional neural network (CNN) model framework, and set parameters to train the model for this challenge. VGG is used as one of the highest-performing CNN models because of its simplicity. Furthermore, the study developed an effective approach to detect brain tumors using MRI to aid in making quick, efficient, and precise decisions. Faster CNN used the VGG 16 architecture as a primary network to generate convolutional feature maps, then classified these to yield tumor region suggestions. The prediction accuracy was used to assess performance. Our suggested methodology was evaluated on a dataset for brain tumor diagnosis using MR images comprising 253 MRI brain images, with 155 showing tumors. Our approach could identify brain tumors in MR images. In the testing data, the algorithm outperformed the current conventional approaches for detecting brain tumors (Precision = 96%, 98.15%, 98.41% and F1-score = 91.78%, 92.6% and 91.29% respectively) and achieved an excellent accuracy of CNN 96%, VGG 16 98.5% and Ensemble Model 98.14%. The study also presents future recommendations regarding the proposed research work.

**Keywords:** brain tumor; Artificial Intelligence (AI); deep learning; convolutional neural network (CNN); feature extraction

## 1. Introduction

The application of information technology and machine learning in medicine has gained importance in today's world. Artificial intelligence is a scientific field concerned with developing a machine that can learn on its own without human intervention to prepare itself for dealing with potential cases on its own. Application of this science finds high relevance in developing interventions for brain tumors, as the tumor cells show highly uncertain behavior that is too complex to be controlled through conventional medicine.

Once tumor cells are generated within the human brain, the likelihood of serious fatalities is created. Due to the complexity of issues, brain tumors are extremely unstable

and potentially fatal in the absence of intelligent solutions [1]. Formation of tumor takes place in the brain during the early stages and can later spread gradually to other elements of the body. To deal with such complex issues, humans can create machines behaving like living beings, capable of learning from experience and applying their experience to cater to the emerging issues due to the accumulation of tumor cells in the brain [2]. In this regard, in the field of medical imaging, AI and digital image processing a huge impact is made by convolutional neural network (CNN) [3].

Some tumors can cause damage to surrounding structures in the brain. So, before performing any brain surgical technique or therapeutic intervention, doctors must specify the exact affected region or area in the brain. Brain tumor segmentation is a process of separating tumors by isolating better and healthier tissues from affected areas. As a result, brain segmentation is the most challenging task in diagnostic techniques. Instead of being specialized in the brain tumor domain, many exclusionary techniques depend on general edge-based data. Due to their efficacy in detecting features of images, deep learning algorithms have lately been used for tumor segmentation tasks [4,5].

A positive brain tumor diagnosis is critical for enhancing treatment outcomes and patients' lives. Radiologists must diagnose brain tumors as early as possible. According to Amin et al. [6], a typical brain tumor can double in size within just twenty-five days. If the person is not treated correctly, the person's survival rate is typically less than 12 months [7].

Keeping the severity of the problem in mind, a fully automated method for detecting brain tumors is required. The manual process of evaluating many images obtained in a clinic is complicated and insufficient to understand the behavior of different tumors. In order to understand and intervene in this complex phenomenon, more precise computer-based tumor detection/diagnosis technologies are required. Several endeavors have been undertaken to investigate machine learning techniques for digitizing this procedure in recent times. Deep learning methods have recently sparked an interest in more accurate and consistent detection of tumor cells.

Automatic defect recognition in medical imaging has emerged as a promising field for various medical diagnostic procedures. The detection and tracking of tumors in Magnetic Resonance Imaging (MRI) is crucial because it offers details about abnormal tissues needed for therapeutic interventions. MRI brain tumor detection is a complicated task, due to the complexities and diverse forms of tumors. Collecting, organizing, and analyzing medical images has become digitized in today's digital realm [8]. Even with cutting-edge technology, thorough interpretation of medical images poses time and accuracy problems. The difficulty is particularly acute in abnormal color and shape areas that radiologists must recognize for future research. There is room in current literature and practice to reap the potential of CNN for image analysis techniques required in brain tumor detection and diagnosis [3]. Considering the demand for advanced machine learning, this research intends to implement the CNN model in light of the current state of knowledge and propose the training of data to handle the complexities arising in detecting brain tumors while offering interventions.

### 1.1. Deep Learning Algorithms (DLAs)

Deep Learning Algorithms (DLAs) are based on a type of AI and machine learning through which imitation of the way humans acquire knowledge is done [9]. When the developments in brain MRI and respective computer interventions are examined, central importance is given to Convolutional Neural Networks (CNNs) and Deep learning (DL).

Conventional neural networks have already progressed or advanced into Deep Neural Network (DNN) approaches. Connections in these networks are data-driven, and the type of methodology is automatically generated without any intervention, which accounts for the precision and impressive performance of these systems in a variety of areas. In reality, it is a deep learning algorithm composed of several nerve-based algorithms that automatically detect features and characteristics in input data before applying the knowledge for developing interventions [9].

### 1.1.1. Convolutional Neural Network (CNN)

AI's potential to bridge the gap between human and machine abilities has grown considerably. To generate outstanding results, both scientists and enthusiasts focus on several subject aspects. The primary concern of this approach is to take control over the machine and to interpret the world the same way that humans do, but then use that understanding for a multitude of jobs, like image processing and computer vision, image processing and categorization, mainstream press entertainment, expert systems, language processing, and so on. Advancements in computer vision have been built and developed, most noticeably through a structured framework known as CNN. CNN is a DL process that requires and prioritizes multiple elements in an image, enabling them to be differentiated from one another. A ConvNet takes considerably less pre-processing than other classification techniques. While simple filtering methods are also effective, with adequate training, they can learn extensions as ConvNet [10].

Individual features in a convolution layer are known as neurons, and the output of a neuron is defined by the pixels around it. The perceptron is the region of pixel density that can influence a neuron's response [11,12]. A neuron's computing efficiency increases as the amount of convolution operation in the design increases. Layers dividing the CNN have a hierarchical system. CNN comprises input, hidden units, convolutional aspects, and so on. It also includes batch normalization and convolution layers. CNN differs in proportion to the number of layers implemented, the size, and the type of activation methods used. CNN variables are empirically determined and empirically supported through trial-and-error. Neurons only react to changes in a part of the area of visuals referred to as the Receptive Field. If similar fields intersect, then they will cover the entire visible region. Figure 1 presents the structure of a single layer Convolutional Neural Network.

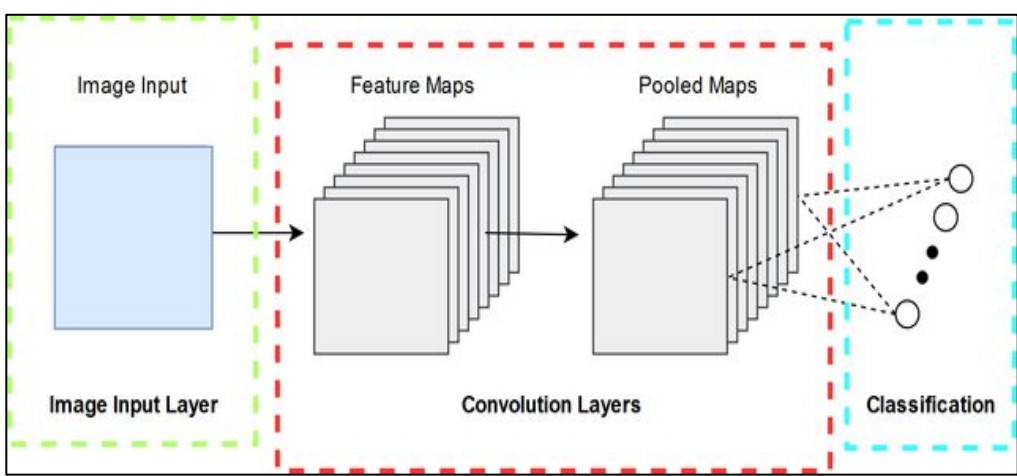

**Figure 1.** Single Layer Convolutional Neural Network.

### 1.1.2. VGG 16

VGG 16 is a model for a 16-layer CNN model. It is still considered one of today's best and most effective models. Instead of having numerous parameters, the VGG 16 model architecture focuses on ConvNet layers with a $3 \times 3$ kernel size. The significance of this model lies in the fact that its values are freely available online and may be downloaded for use in one's systems and applications. When compared to other developed comprehensives, it is noted for its simplicity. This model's minimum expected input image size is $224 \times 224$ pixels with three channels. In neural networks, optimization algorithms are used to evaluate whether a neuron must be engaged or not, by determining the weighted sum of input. The need for kernel function arises from inducing non-linearity into the output neuron. A neural network's neurons function together with weight, bias, and the related training procedure. The neurons' link weights are adjusted based on the output inaccuracy. The input layer and the activation function add non-linearity to artificial neural

input, allowing it to learn and accomplish complex tasks. Figure 2 presents a standard VGG 16 network architecture.

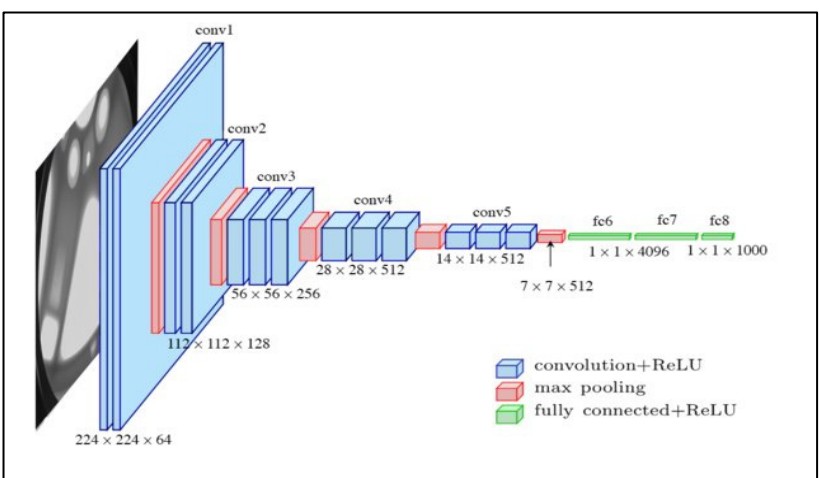

**Figure 2.** A Standard VGG 16 Network Architecture.

### 1.1.3. Ensemble Model

For many machine-learning challenges, Ensemble modeling emerges as a state-of-the-art solution, as it aggregates the predicting capacities of multiple models such that higher performance can be achieved. Different training data sets or modeling techniques are utilized together in Ensemble modeling which further integrates the forecasts of each base model, to give out a single predictive performance for the unknown data. The use of ensemble models intends to diminish forecast generalization capability. With multiple models, the prediction error decreases as long as base models are diversified and independent. Deep Learning Synthesis seeks to improve efficiency by merging features from different models into a unique predicted feature. In addition, ensemble learning can be classified into data ensembles, and classifiers depend on the scale of integration. Since the feature set includes more data about the MRI images than all classifiers combined, incorporation at this level is expected to improve classification performance. The classifier ensemble consists of classifier output sets, wherein voting methods determine output. In contrast, a features ensemble consists of extracted features given for the classification for the final result.

In these models, the probabilities are estimated by beginning an active contour and executing it until a likelihood smaller than the stated threshold is detected, to give out an indication of the tumor area [13,14]. As matching a big tumor area to an image is a complex process, some algorithms in literature have included atlas registration, in contrast to tumor segmentation [15]. The specified critical values of the proposed model would imply that it successfully located the primary tumor region while avoiding false positives. Most existing practices refer to the entire tumor site, which results in low core performance metrics and attempts to improve regions. These limitations stress the importance of the traditional architecture proposed in this work. In the research, methods contain both computational and scientific processes [16]. Tumors, by definition, can form in any spatial location in the brain and have a unique and irregular structure, making automated methods challenging to detect. Due to the fact that most previous strategies were tested on manually produced datasets, generality cannot be assured. The brain tumor diagnosis computation and computer-assisted therapy society gathers, disseminates, or establishes MRI image datasets, comprising glaucoma tumor MRI scans [17,18].

In this study, researchers investigated and examined the effectiveness of two machine learning techniques, CNN and Bi-LSTM. In addition, the study proposed using ML techniques to overcome the shortcomings of classical approaches. Since these ML algorithms have been shown to perform effectively in most pattern categorization problems, they are helpful as neural networks can learn to produce intricate mappings between input and

output. They can handle far more difficult classification and detection problems than those at hand.

In a nutshell, the CNN results on the data are used to get an optimal classification approach of improving the previously proposed methodologies. A deep learning algorithm (Bi-LSTM) would be used to extract features. The clustering method is used and applied to a data collection. The results would assess the efficiency of the recommended technique. Using the feature before CNN can lead to higher picture classification results, and using Bi-LSTM would increase network accuracy. Figure 3 presents the architecture of the ensemble model.

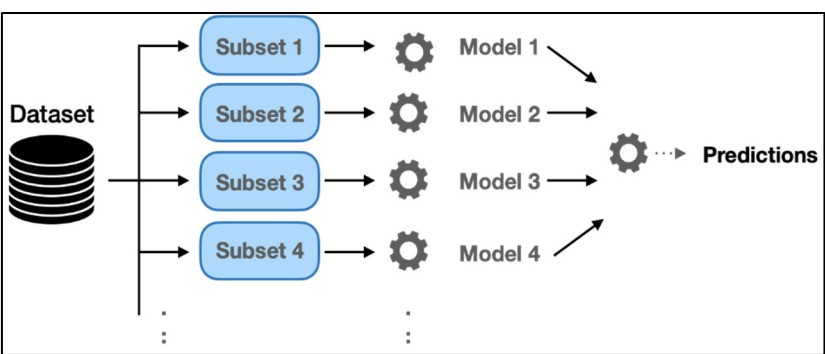

**Figure 3.** Architecture of Ensemble Model.

### 1.2. Research Objectives

The proposed technique for identifying and classifying tumors from brain scans and images used CNN and DL techniques. These networks are constructed from neurons with learnable weights and biases. The proposed study aimed to critically analyze how researchers solved brain tumor issues in previous literature by using Visual Geometry Group (VGG 16) to discover a brain tumor, implement a CNN model framework, and set parameters to train the model for this challenge. VGG was used as one of the highest-performing CNN models because of its simplicity. Furthermore, the study developed an effective approach for detecting brain tumors using Magnetic Resonance Imaging (M.R.I.) scans for tumor detection to aid in making quick, efficient, and precise decisions, and conducted segmentation of the data sources we intended to use in the proposed research work.

### 1.3. Strength and Novelty in Proposed Model

The novelty of the study is a unique solution being proposed by the researchers that can assist with the detection of brain tumors in the acquired MRI image, offering great precision. The proposed model deploys a technique that intends to offer training for models with higher optimization compared to previous techniques in the literature [19–28].

High optimization of training would reduce required computational power, marking our model's significance for practitioners and researchers. The study model comprised a combination of deep learning and transfer learning models to facilitate the achievement of a remarkable accuracy rate that would exceed the performance of other available solutions.

## 2. Literature Review

A group of irregular brain cells make a brain tumor. Ambiguous development in the synapses characterizes brain tumors. The high rigidness and inflexibility of the skull makes it problematic to contain potential expansion in volume. The impact may be experienced in the form of interference with human capacities along with swelling in to other body parts [29]. Over 130 different brain, and central nervous, cancers range from benign to malignant, exceedingly rare to reasonably common [29]. However, each of the 130 brain tumors is classified as either primary or secondary. Moreover, the preponderance of brain

malignancies are secondary brain tumors [30]. Breast, stomach, or skin cancer begins in another portion of the anatomy and progresses to the brain.

In literature, machine learning, deep learning in particular, has been argued to have the potential of overcoming the challenges associated with the detection and intervention of brain tumor [31]. Deep Learning is regarded as one of the best methods in data science and artificial intelligence to train models through data to develop valuable decision-making abilities [32]. Obtaining the predicted network by reducing the image without sacrificing the information required to make predictions is sought in such models. Segmentation of tumor is known to be best performed by deep learning techniques, making it a viable choice for implementation [33].

Deep Learning [9], which resembles the tasks of the human brain, is one of the operations of Artificial Intelligence. It is being used to detect computational artifacts, recognize the voice, translate language skills, and make decisions. It may comprehend without human management, demonstrated by unorganized and unlabeled data. CNNs are deep neural networks, often utilized in deep learning visual depiction analysis [15]. The rest of this section is dedicated to exploration of current literature for application of deep learning in medical diagnosis.

Viewing the current literature, one cannot deny that Computed Tomography (CT) brain scanning is one of the most common applications of deep learning, with learning algorithms producing the most accurate results [34]. Current research introduces new connectivity supervised learning, LM for deep CNN topologies that combine (CNN) and classic architectures. The implicit condone neural network assists CNN in identifying the optimum files for aggregate and convolution. As a result, the primary neurological classification algorithm [35] learns faster and more efficiently. In current study, researchers used brain MRI images from the Kaggle portal [36], created for brain tumor categorization research. The collection comprises 138 images of people who are perfectly well and 200 scans of sick people. Images come in a variety of sizes and formats.

In the study of Marcin Woźniak, Jakub Siłka and Michał Wieczorek [37], a DL-based supervised technique for detecting changes in synthetic aperture (SAR) images was proposed. This method generated a suitable volume of DBN data using MRI images employing a segmentation technique. This method's detecting performance suggests the applicability of deep learning-based approaches for handling change identification problems. Moreover, the study describes an automated brain tumor classification system based on DNN [38]. The suggested networks are intended to be used in images of glioblastoma illness. A new convolution layer is proposed in current research and the suggested cascade design uses the results of a core CNN as an incidental finding for the following CNN.

Categorizing glioma tumor imaging can be incredibly useful in computer-aided diagnostics (CADs); however, it is a time-consuming process. Each MRI has 150–220 brain tumor images produced at healthcare centers, and regularly grows. Doctors assess a tumor individual using approximate value and instinct, threatening the patient's life [39].

A machine learning-based technique for segmenting brain tumors using several MRI modes is provided in this paper. The suggested hybrid CNN architecture predicts output labels using a patch-based method that considers local and factual data [40]. The proposed network handles the over-fitting problem by combining dropout regularization with the compound considered, while a two-phase training technique addresses the issue of the varying definition. The proposed method incorporates a pre-stage in which images are normalized, and the bias field is modified, a CNN feed-forward run, and the elimination of results encompassing the skull region. The research methodology is validated using the BRATS 2013 data, yielding dice scores, sensitivities scores, and specificity scores of approximately 0.86, 0.86, and 0.91, respectively.

MRI is perhaps the most often used method for imaging brain regions. Since MRI moderates tissue differences, it is the most versatile imaging method for simulating brain regions of interest, such as tumors [41,42]. The primary purpose of tumor segmentation is to identify and exact the segmentation of metastatic tumor voxels, like edema, necrotic

centers, and tumorous tissues. DL techniques develop standard NN by including hidden layers to the network structure between output units to reflect more complex and nonlinear connections.

Although the network model does not need feature extraction to apply a small amount, training CNN architecture is complex and difficult since it needs a dataset for testing and training before the structure is ready for classification, which is not always available. In addition, hardware is required for computing the massive factor for large image sizes [43,44].

The proposed approach for detecting brain cancers is based on deep learning, and brain magnetic resonance is used for tumor evaluation using automated classification methods. For image improvement, triangular fuzzy median filtering is used, which aids in reliability, such as for the unsupervised fuzzy set approach [45,46]. Gabor characteristics are extracted from each individual's lesions, and similar texture (ST) features are derived. These ST traits are fed into transfer learning (ELM), including one for tumor categorization. The proposed strategy produces better results while consuming less processing time. The suggested system of the current study is divided into three sections: augmentation, image pre-processing, and Convolutional Neural Networks (CNNs). The strategy suggests a methodology that uses an asset and a deep learning algorithm. The CNN has an 87.42% accuracy percentage with minimal difficulties involved, distinguishing it from all other strategies [47].

The hidden layers differentiate CNN models. Two convolution layers use termination criteria and batch normalization for regularization in addition to the dropout layer. Without such layers, the other two models are used. For training, the BRATS 2013 Dataset was used, while for testing, the World Brain Atlas was used (WBA). According to the data, model 4 provides a satisfactory false alarm rate for non-tumor images, whereas model 6 provides the desired results and achieves 96% accuracy [48]. Since DL Algorithms can effectively express complex interactions without needing a wide variety of equipment, they are a remarkable development in ML (KNN) [49,50]. As a result, they evolved swiftly to become the cutting-edge in several health informatics fields, such as informatics, healthcare analytics, and pattern classification.

## 3. Comparative Analysis

The research needed to look into the most recent cutting-edge studies on brain tumor identification and tracking. This study evaluated recent papers published in the last decade or so that focused on the identification and categorization/classification of brain tumors employing CNN and VGG 16 algorithms.

*Existing Methodology*

Current framework systems follow multiple pre-defined procedures to identify brain MRI images [51–54]. The effective mechanisms involved in recognizing and classifying tumor and non-tumor units in MRI brain imaging is covered in [55,56]. The following is a brief overview of prospective approaches and strategies. Most images to be entered as input are MRI brain scans [57,58]. Depending on the architecture and memory limitations, the input might be 2D or 3D. Due to its efficiency in significantly enhancing image data the input regarding images to be entered has proved to be as crucial as any other stage [19–21,59].

Segmentation primarily divides the input image into identical sections depending upon specified criteria, allowing only essential data to be extracted and the remainder to be discarded [22,23]. There are numerous approaches. Some studies segment the actual tumor [48], whereas others segment the image region including the tumor [25]. The goal of the classification stage is to divide the input data into various categories based on comparable behavior patterns inside the group.

The third process described in the literature involves directly feeding brain MRI images to a Deep Learning program for categorization with no pre-processing. Statistical techniques or machine learning techniques are used to identify features. The Deep Learning algorithm is then trained using these extracted features. While deep learning methods do

not necessitate extraction of features, the study has shown that extracted features using machine learning, or meta-heuristic optimization methods, are still used in different models with reinforcement learning to include efficient and resilient features [27].

Each plan's principal purpose is to change the levels of Supervised Learning based on the experimental criteria and then choose the model with the best performance. Using machine and Deep Learning approaches, researchers employed models to build efficient systems. The dataset is divided into learning, testing, and verification sets before beginning any of the approaches above. Convolutional Neural Network (CNN) has received substantial appreciation and recognition in Deep Learning for its ability to automatically extract and detect deep features by responding to tiny changes in images.

A constant comparative table (Table 1) briefly describes all of the essential characteristics of each previously presented study paper in this area. Table 1 essentially provides a concise overview of the tactics employed up to this moment. Some limits and inadequacies, as well as the obtained results, have been highlighted for thorough analysis.

**Table 1.** Comparative Analysis.

| Ref. | Year | Methodology/ Approach | Dataset | Result | Drawback |
|------|------|------------------------|---------|--------|----------|
| [60] | 2020 | CNN | Fig-share Total Images (3064) from 233 patients) | 87% and 92% | No training time has been mentioned |
| [51] | 2019 | PNN Classification CNN | KaggleTCIA | 90% Accuracy | Lack of comparative analysis |
| [52] | 2019 | R-CNN And SVM | Pvt Dataset | 95% Accuracy | Rapid conversion convolutional feature map into the region proposed |
| [53] | 2020 | Inception Pre-trained CNN | BRATS 13,14,17,18 | 92% Classification | Complex approach |
| [54] | 2019 | ResNet-50 for Detection GAN for Data Augmentation | BRATS 2016 | ResNet87% accuracy With GAN, 92% | Certain major aspects are not mentioned |
| [55] | 2019 | CNN | A private dataset comprising 330 images | Accuracy 98% with low Complexity level | Lower-level data implementation. |
| [56] | 2019 | CNN | Fig-share dataset (3064 images) | 96% Accuracy | Lacking comparative analysis |
| [57] | 2019 | 3D-Multi CNNs | BRATS 2018 | Coefficient 84% Sensitivity 82% Specificity 99% | Ambiguous results and no exceptions carried out |
| [61] | 2019 | Alexnet, VGG-16 | BRATS 2015 | VGG16 gives 98% accuracy | There is no information about enhancement other than normalization. |
| [56] | 2016 | D.N.N. and ELM | 1000 Images private Dataset f om some Indian hospital | D.N.N. showed 88% accuracy E.L.M. Delivered 96% Accuracy | Complex features extraction process |
| [59] | 2017 | Convnet and slicenet and VGNet | BRATS 2017 And TCIA | 97% Accuracy being achieved | Scheme + high training time |
| [19] | 2017 | 3D CNN | BRATS 2015 | 75.4% for Flair 1.3% improvement 74.2% with 3.3% improvement | Limited data on studies previously done |

**Table 1.** *Cont.*

| Ref. | Year | Methodology/ Approach | Dataset | Result | Drawback |
|---|---|---|---|---|---|
| [20] | 2018 | Classification D.N.N. and Segmentation Using Fuzzy C | Harvard Dataset (66 M.R.I.s with 22 and 44 standard images vs. affected. | 98% | The methodology used was not novel. |
| [21] | 2018 | CNN for Classification CNN for grading | data sets Fig-share 3064 and REMBRA NDT 516 images) | Accuracy of 96% and 98% | High learning rate |
| [22] | 2020 | BRAIN NETs For detect And Classification | (BITE), Fig share (4689 detection and for classification) | 98% accuracy For detection and 99% | Complex architecture |
| [23] | 2020 | BRATS, CE-MRI | VGGnetAnd KNN as classifier | 97.28% and 98.69% on Both of the datasets | No training testing time, |
| [24] | 2020 | R.N.N. | Private dataset comprising 1000 images | Classification 96% Specificity 98% Sensitivity 97% | There was no specific place. This process is examined. |
| [25] | 2019 | CNN | Kaggle | Accuracy 92.3% | Basic model |
| [26] | 2020 | ELM- LRF CNN | Figshare dataset (3064 images) | 97% | Small training data. |
| [27] | 2020 | CNN | Kaggle | 90 to 99% Accuracy | No preprocessing data |
| [28] | 2019 | Residual Network ResNet | Fig-share (3064 images) | 95% Accuracy being achieved | Ambiguous data depiction |
| Proposed Model | 2022 | CNN, VGG-16, Ensemble Model | MRI dataset | 96%, 98.15%, 98.41% | High accuracy |

Similarly, no efficiency statistic constitutes a universal standard for all studies. The researchers used a set of performance assessments, but the interpretations and mathematical depiction of performance indicators in Table 1, and the findings in Table 1, were decisive. They are all related to the accuracy of parameter estimates in some way. When it comes to detection, excellent accuracy and performance are essential.

As a result, it was found that the critical concern of the specialists remained accuracy. However, we know that MRI scans have an imbalanced class issue, so that that accuracy would not help much. Various ML techniques are available that can improve model performance despite imbalanced data, such as focal loss techniques, SMOTE (Synthetic Minority Over-sampling Technique), and class-weight [62]. When there is a class discrepancy, precision, and recall work well. The majority of the research used these in combination with accuracy. Precision ensures correctness, while recall indicates if the minority class, or the values that guide, were covered. In this perspective, both indicators are equally important in terms of accuracy.

## 4. Proposed Research Methodology

The study aimed to automate the detection of brain tumors in MRI brain scans. Our suggested strategy uses CNN and VGG 16 to detect brain tumors employing brain MRI data. According to Devi [63], VGG 16 can be considered a contender for other optimization methods, such as AlexNet and Grid Search optimization, while detecting and classifying brain tumor MRI with CNN. Considering the rationale of [63,64] for the application of AI in diagnostics, the proposed framework was broken down into many steps. The brain MRI image was used as the primary input image. Data operations, including thresholding and refractive error, were carried out to reduce noise. The database of brain MRI images was analyzed and improved. The images were then resized for use as input to the model.

A VGG 16 pre-trained convolution layer was used to improve and increase classification accuracy into two classes: yes and no. VGG 16 is a significantly improved performance VGGNet variant among the most advanced classification networks. Figure 4 depicts the structure of the proposed approach.

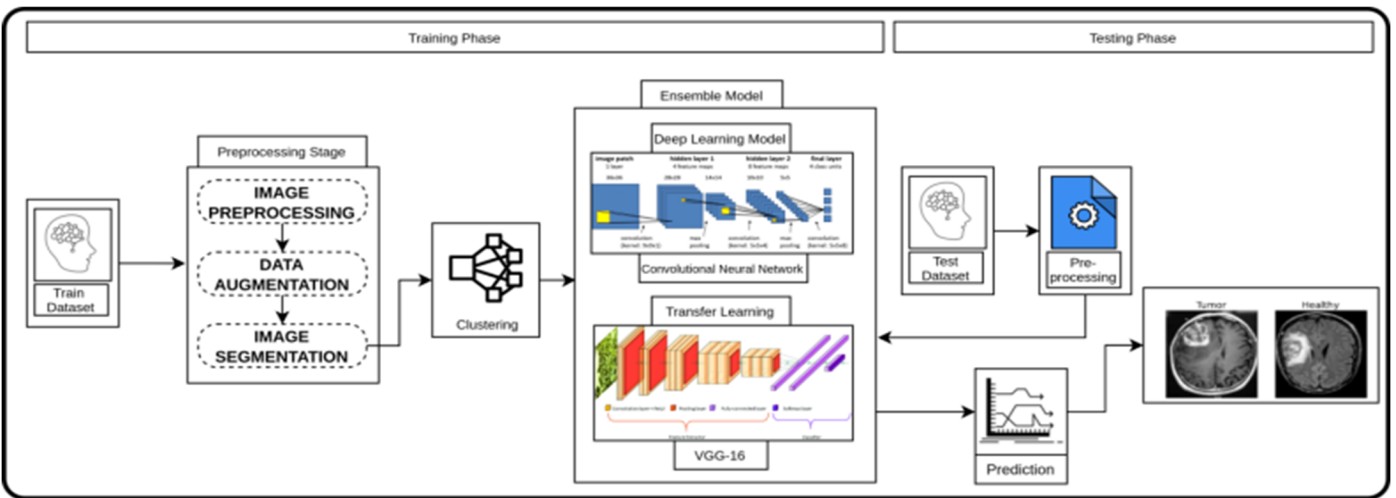

**Figure 4.** Proposed Approach.

### 4.1. Materials and Methods

An automatic brain tumor detection and classification method were implemented in this research using the Faster CNN algorithm. Faster CNN used the VGG 16 architecture as a primary network to generate convolutional feature maps, then classified these to yield tumor region suggestions. Prediction accuracy was used to assess performance.

To detect brain tumors, several researchers use development tools, such as MATLAB. However, the proposed research work opted to use Python programming to complete our project objective. Among the reasons the paper chose Python are:

- Python is free and open-source, with more datasets and graphical packages than MATLAB [63].
- Python code is much more precise and accurate than MATLAB code. Python provides greater control over implementation to achieve and better name visual-spatial skills.
- We could easily manage a variety of classic library versions.

### 4.2. Data Pre-Processing and Dataset Division

Due to potential disturbances that are incurred while obtaining MRI machine boundaries, MR images may accrue discrepancies, such as inhomogeneity deformations and the heterogeneous nature of the movement. These artifacts cause false intensity rates to be induced, resulting in false-positive results in the image. To collaborate with these artifacts, the N4ITK system corrects bias field distortion.

After approaching the MRI image, a pre-processing procedure was initially applied. CNN struggles to adjust to the peculiarities of individual classifiers because the magnitude in MR images have irregular black edges. Amplitude normalization was used to narrow the intensity distribution to a normal range, culminating in a mean intensity value of zero and a standard variance of one. To begin, the images were thresholder at 45 to remove any minor patches of deformation, followed by a series of deprivations and dilations. The photos were then normalized by gathering the most extensive contours of each image and cutting the images on the contour's excess top, bottom, left, and right ends.

### 4.3. Dataset

The dataset used to develop a respective framework in this study was "Brain MRI Images for Brain Tumor Detection." The employed dataset included three distinct and well-known kinds of brain cancer: meningioma, glioma, and pituitary tumors. Sted models were trained and tested using an MRI dataset that included 253 brain images from 155 different patient features and cases. The data tumor dataset was submitted for processing. Figure 5 depicts the structure of the Brain MRI Images dataset sample and classification process, as discussed below.

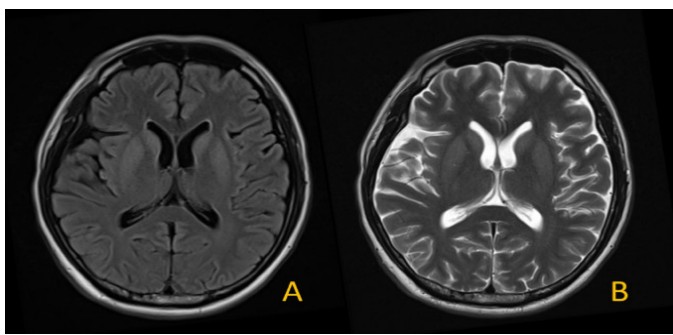

**Figure 5.** Brain MRI Images Dataset Sample (**A**) = image with dark margins, (**B**) = image without dark margins).

### 4.4. Image Processing and Classification

This study focused on using convolution to eliminate the dark margins from the images before extracting only the brain region from MRI data. The detection technique was a multi-phase strategy for detecting the edges of objects in images. The edges of the Real MRI brain have been shown using a clever edge detection algorithm, and then only the brain section of the image was trimmed.

Image enhancement boosts network performance by purposely creating more training data from original data. The input image of the VGG 16 Network had a size of 224. Thus, the training dataset was used to resize our data and make it more suitable for the process of classification. It is a procedure that is used throughout DL to aid in the creation of samples. It also optimizes the network's efficiency for a relatively small dataset. The images were altered to add variance to account for the dataset's small size. As a result, image extensions were used to increase the variability within our constrained dataset by collecting Keras Image Data Generator when training. Figure 6 depicts different types of the brain tumor and shows the classification of the dataset into two different classes, A and B.

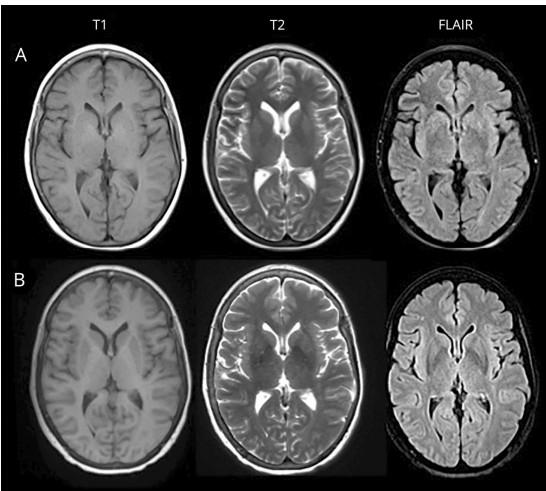

**Figure 6.** Brain tumor types. (**A**,**B**) = two different classes of the dataset.

The entire augmentation process was as follows. The photos were shifted at random angles ranging from 0 to 15 degrees (clockwise). These were shrunk to 10% of their original size and shape. The visuals were arbitrarily boosted or dimmed from 0% to 50%. The images were also sheared at an angle of 0.1 radians (counter-clockwise). Finally, the photographs were randomly rotated horizontally and vertically.

*4.5. Implementation Details*

Convolutional Neural Network (CNN) and VGG-16 Network Framework

- CNN

As a result of the hidden possibilities of using the geometry of the images, CNN's primary applications are in photo editing. In graph analysis, CNN outperforms numerous techniques. Various architectural concepts are combined: receptive fields, batch normalization, and spatial or temporal sub-sampling. Figure 7 depicts the structure of the suggested CNN model.

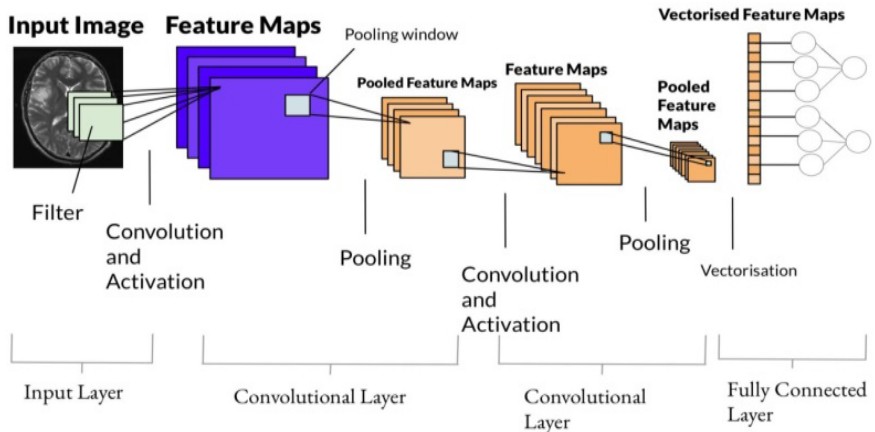

**Figure 7.** Flow Diagram of CNN Model.

For tumor identification, a multi-layer convolutional neural network was constructed and implemented. An input shape of $64 \times 64 \pm 3$ was built for the MRI scans using a convolutional layer as the start layer, converting all of the pictures into a homogeneous dimension. After collecting all of the images in the same aspect, we developed a convolution kernel entangled with the input layer. We managed 32 convolutional filters of size $3 \times 3$, each with the assistance of 3 channel tensors. The cumulative model, which consisted of seven steps, including the hidden layers, produced the most precise result for malignant concerns. ReLU did not relate to the output since it was used as an activation function. ReLU could be stated mathematically as,

$$F(x) = \max(0, x) \tag{1}$$

Processing the MRI image of the brain could lead to overfitting contaminants. We used MaxPooling2D to generate spatial database models based on the input data. This convolutional layer had size $31 \times 31 \times 32$. The pool size was reduced due to splitting the input photos in both spatial directions (2, 2).

$$(z)_x = \frac{ae^{zi}}{\sum_i^l ae^{zi}} \tag{2}$$

The metrics used to analyze the study's outputs included specificity, sensitivity, f-score, and accuracy. These metric values were computed using the confusion matrix. As a

consequence, the CNN model parameters were true positive (TP), true negative (TN), false positive (FP), and false-negative (FN).

$$\text{Accuracy (\%)} = (TP + TN)/(TP \; FP \; TN \; FN) \tag{3}$$

However, as previously stated, accuracy, sensitivity, or recall, as well as the F1-score values, were also considered when examining the CNN performance of the model. These metrics' equations are:

$$\text{Precision (PPV)} = TP/(TP \; FP) \tag{4}$$

$$\text{Sensitivity} = TP/(TP + FN) \tag{5}$$

$$\text{F-score} = \frac{2(\text{Percision} * \text{Recall})}{\text{Percion} + \text{Recall}} \tag{6}$$

$$P_0 = \frac{TP + TN}{TP \; FN \; FP \; TN} \tag{7}$$

$$P_{Yes} = \frac{TP + TN}{TP + FN + FP + TN} \cdot \frac{TP + FN}{TP + FN + FP + TN} \tag{8}$$

$$P_{No} = \frac{FP + TN}{TP + FN + FP + TN} \cdot \frac{FN + TN}{TP + FN + FP + TN} \tag{9}$$

$$P_e = P_{Yes} + P_{No} \tag{10}$$

- Ensemble Classification

The Ensemble is a machine-learning method that provides numerous basic models to generate an optimized predictive model. Many ensemble approaches have been identified in the literature; however, the method with the highest number of votes was chosen for this work. In most categorization situations, the majority technique was used. To produce recommendations for each data point, this strategy employed numerous models. Predictions for each model were taken into account. Most models' forecasts were used as final predictions. The suggested technique employed a Convolutional Neural Network (CNN) and a VGG 16 ensemble model. The characteristics were obtained from the VGG 16's final fully-connected layer.

**5. Results**

The fundamental goal of our suggested study was to develop a well-fitting model, while eliminating underfit and overfit issues. We concluded that our model did not induce overfitting or underfitting. When comparing training and test data, the model loss should be lower in training data. We discovered that deep learning systems, such as tensor flow and Keras, were advantageous when using neural nets to solve classification tasks.

The benefit was that if we understand these fundamental notions and how effective convolutional networks are, we can handle even the most challenging problems. Model loss in training examples should be smaller than in test data. We found that deep learning frameworks, such as tensor flow and Keras, were preferable when employing neural nets to perform classification tasks. Convolutional networks could tackle the most challenging problems if we comprehended these core concepts and the learning curve. These powerful adjustment options could help us avoid fitting issues. The generalization difference was the steeper learning curve gap between the training and test loss.

$$\text{Accuracy} = \text{Number of correctly predicted images} \tag{11}$$

$$\text{Total number of images} \times 100 \tag{12}$$

*5.1. Training and Validation Accuracy VGG-16*

Accuracy and loss of the VGG 16 Model during training and validation were assessed. Training indicated that AUC was not constant and became increasingly nonlinear as the

number of repetitions increased. The validation of AUC in VGG 16 remained unchanged at 98.15%. We employed two classes and evaluated the accuracy of using the above approach. Our model achieved an f1 score of 92.6% with a recall of 94.4%. Our system improved when the number of trained images and hyper-parameters increased. Figures 8 and 9 present the training and validation loss of VGG 16, and training and validation accuracy of VGG 16, respectively.

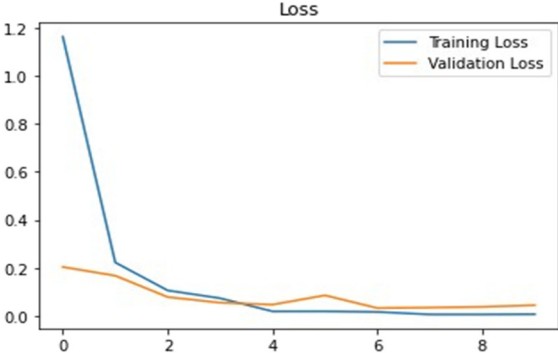

**Figure 8.** Training and Validation Loss VGG-16.

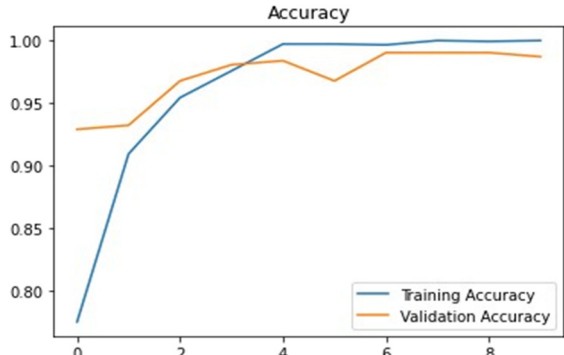

**Figure 9.** Training And Validation Accuracy VGG-16.

*5.2. Training and Validation Accuracy CNN*

The training indicated that recollection was inconsistent and very nonlinear as a repetitions approach. Figures 10 and 11 present the training loss and accuracy of CNN and training and validation accuracy of CNN, respectively.

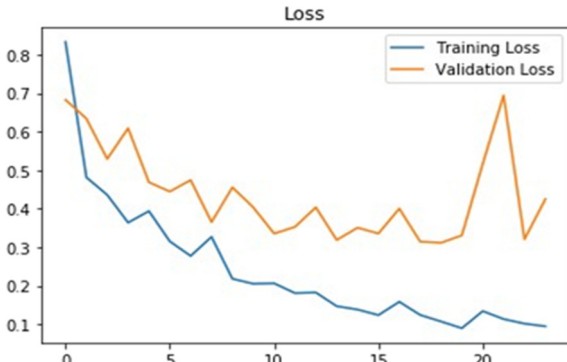

**Figure 10.** Training Loss and Accuracy CNN.

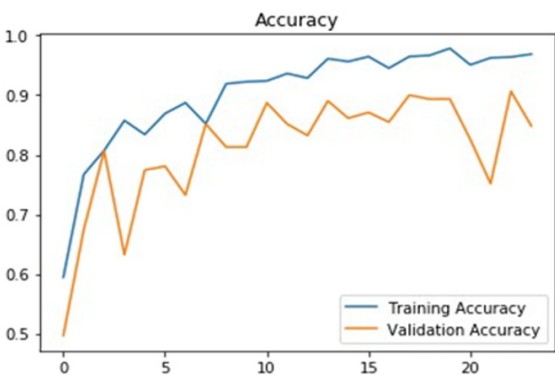

**Figure 11.** Training and Validation Accuracy CNN.

According to the training, the recall metrics measurement nearly reached a change in iteration and remained virtually constant as the number of iterations increased. The generalization difference was the steeper learning curve gap between the train and test loss. The advantage is that if we comprehend the core concepts and the learning curve, convolutional networks could tackle the most challenging problems. These powerful adjustment options could help us avoid fitting issues.

### 5.3. Ensemble Model: Validation and Training

Compared to VGG 16, which obtained a training and testing accuracy of 98.15% and recall of 97%, the suggested ensemble model achieved a training and testing accuracy of 98.41% with an F1-score of 91.52% and recall of 91%. Compared to loss performance metrics and ensembles, the model testing loss was too low at roughly 2.01 correspondingly. Figure 12 presents the training and valuation accuracy of the ensemble model.

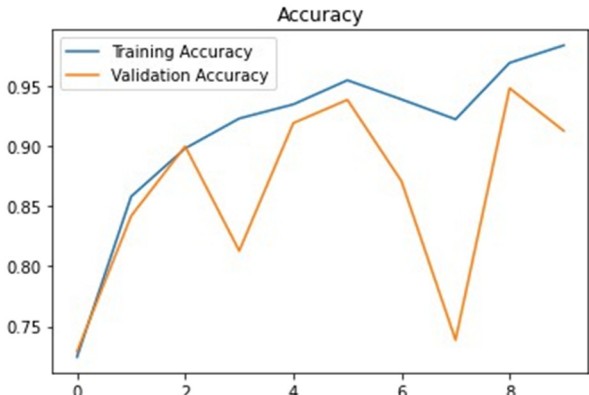

**Figure 12.** Training and Validation Accuracy Ensemble Model.

### 5.4. Result Analysis and Discussion

We ran our tests on a PC running the Linux 4.0.0 LTS platform with a Geforce R.T.X. 2080Ti GPU. On an Intel Xeon-2620, Core i5-2.4GHz CPU, and 16 GB RAM, the CNN VGG 16 and Ensembles models were verified using Python in the Keras module. The CNN, VGG 16, 80% data was reserved for training, 10% for validation, and the other 10% for testing, a learning rate of 0.0001 over 80 epochs, a batch size of 16, and category cross-entropy as the error rate were used to train the Network.

Our suggested framework comprised CNN, VGG 1, and Ensemble, respectively, with reasonable accuracy achieved via CNN at 96%, VGG 16 at 98.5%, and Ensemble Model at 98.14%, as described in Table 2.

**Table 2.** Performance and Result of Proposed Approach.

| Models | Accuracy | Recall | F1-Score | Validation_Accuracy | Test_Accuracy |
|---|---|---|---|---|---|
| CNN | 96% | 89.5% | 91.76% | 87.34% | 89.5% |
| VGG-16 | 98.15% | 94.4% | 92.6% | 98.01% | 97.6% |
| Ensemble Model | 98.41% | 91.4% | 91.54% | 91.29% | 91.29% |

## 6. Discussion

As seen in the preceding research investigation, the acquired accuracies for brain MRI classification using deep learning approaches are much higher than those obtained using classical ML techniques. On the other hand, deep learning algorithms require enormous quantities of data for training to surpass traditional methods. According to recent studies, deep learning approaches have obviously crossed the threshold of specialized and intelligent systems and computer vision. Furthermore, the approaches have limits that should be considered when working with brain tumor diagnosis and classification. So, in Table 1, our proposed solution was compared, in terms of performance, with the previous research to show how improved performance could be achieved with the integration of deep learning and transfer learning models. Comparison with past literature established the need for this study, while arguing for higher performance achieved through implementation.

This study made an attempt to meet such needs by bringing a novel solution for the detection of brain tumors in MRI images with greater precision. The proposed model integrated deep learning and transfer learning models to achieve a remarkable accuracy rate. As compared to similar techniques from past literature, in this study, the optimization of training models was increased to reduce the need for high computational power.

Early diagnosis and suitable viable treatments are essential to adequately treat brain tumor diseases. Alternative treatments are defined by tumor stage, pathological type of disease, and tumor stage at the initial diagnosis. The study results gained in this research were contrasted to state-of-the-art approaches in the base tumor detection challenge or standard deep learning-based techniques that have been offered. Conventional recognition systems use some fundamental machine-learning-oriented methodologies that collect only low- and high-level characteristics in the early phases of critical extracting features. Therefore, Table 2 shows how the suggested hybrid design outperformed approaches in all brain tumor types and categories. The suggested method was particularly effective for the central and boosting regions and improved tumor detection cells, resulting in high sensitivity estimates for the respective areas. The proposed model had an accuracy of 89%, which could be enhanced using other techniques. Implementing image processing techniques and evaluating other AI techniques' effectiveness could achieve the same results. Soon, several organizations and business models are to be proposed to assist radiologists and physicians in the quick and adaptable detection of brain tumors using AI.

The essential variables stated for the proposed model demonstrated that it was efficient in finding original tumor areas, while avoiding false positives. Most current approaches concentrate on the entire tumor region, resulting in poor performance measurements for core and augment regions, emphasizing the importance of the design developed in this study. The methods presented in the literature should include statistical and deep learning-based methodologies, with CNN excelling at dealing with task complexity. Due to the clinical importance of the tumor detection problem, time constraints, sensitivity, and effectiveness are essential. The results validated the efficiency and efficacy of the proposed approaches, especially in terms of fundamental and augmenting regions and specific values, where it outperformed previous methods by a wide margin.

## 7. Conclusions

The diagnosis of brain tumors is essential in clinical treatments. It is critical to interpret medical images because medical images vary greatly. The automatic brain tumor detection approach makes detection easier, but it also significantly increases the patient's chances

of survival. Convolutional networks for brain tumor categorization have helped pave the way for better tumor detection and accuracy. MRI is most commonly used to detect and classify brain cancers. Due to the apparent efficient feature extraction capacity of DL-based techniques, they have recently gained greater attention and efficiency when compared to standard classification techniques for medical imaging. If cancer is diagnosed, many lives can be spared, and the appropriate grade is determined using quick and low-cost diagnostic tools. As a result, there is an urgent need to create rapid, non-invasive, and cost-effective diagnostic tools. This study made an attempt to meet such needs by bringing a novel solution for the detection of brain tumors in MRI images with greater precision. The proposed model integrated deep learning and transfer learning models to achieve a remarkable accuracy rate. As compared to similar techniques from past literature, in this study, the optimization of training models was increased to reduce the need for high computational power.

In this study, a CNN was built to detect brain tumors using MRI scans of the brain automatically. The network could be trained for faster and more convenient training using a pre-trained VGG 16 model. VGG 16 has sixteen layers and is a critical CNN model to evaluate if employing a commercial model for a task. This paper aimed to discover a brain tumor using the VGG 16, CNN model architecture, and weights to training data. The precision of the outcome was evaluated. Brain MRI images for tumor identification are the type of data we aimed to acquire for our study. Compared to conventional methods, the results showed that the proposed network architecture was appealing and performed exceptionally well in detecting tumors. Various processing operations were also carried out to enhance the model's efficiency.

The proposed approach for brain tumor diagnosis was based on deep learning, and brain magnetic resonance was used for tumor evaluation using automated classification methods. For image improvement and better classification, a CNN model was used in this paper. The study successfully yielded better outcomes while requiring less computing time. Our method was used to identify brain tumors in MR images. The algorithm significantly outperformed previously studies. The methodologies used for detecting brain tumors in the testing data (Precision = 96%, 98.15%, 98.41%, and F1-score = 91.78%, 92.6%, and 91.29%) achieved high accuracy of CNN 96%, VGG 16 98.5%, and Ensemble Model is 98.14%. The reliability of the validation and learning was discovered to be increasing, and the findings higher. They could be utilized to diagnose the existence of a tumor in the brain.

The limitation of the research lies in its exclusive focus on brain tumors that can be extended to account for different types of cancer attacks in MRI images. Future research can make use of a variety of image modalities and diverse segmentation techniques to acquire the best approximation of affected regions in the brain to isolate these regions from unaffected parts of the brain. Different modalities having image registration distinctions from each other could be utilized for presenting the missing image features in the fixed image and conducting the best classification. To achieve higher precision and accuracy, ensembles could be further used.

**Author Contributions:** Conceptualization, C.O.N.; Data curation, A.Y. and M.J.A.; Formal analysis, A.Y., L.Q. and M.J.A.; Funding acquisition, L.Q.; Investigation, A.Y., L.Q., C.O.N. and M.J.A.; Methodology, A.Y.; Project administration, A.Y. and H.B.K.; Resources, C.O.N.; Software, A.Y.; Supervision, L.Q. All authors have read and agreed to the published version of the manuscript.

**Funding:** This research was supported by the National Natural Science Foundation of China under Grant No.61471263 and No.61872267, the Natural Science Foundation of Tianjin, China, under Grant 16JCZDJC31100, and Tianjin University Innovation Foundation under Grant 2021XZC-0024.

**Institutional Review Board Statement:** Not applicable.

**Informed Consent Statement:** Not applicable.

**Data Availability Statement:** Not applicable.

**Conflicts of Interest:** The authors declare no conflict of interest.

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
