# Peer review of "Brain Tumor Analysis Using Deep Learning and VGG-16 Ensembling Learning Approaches"

_applsci, doi:10.3390/app12147282_

Round 1
Reviewer 1 Report
I commend the authors about the great lengths they have gone to explaining the issues, needs and outlining their study. However, the manuscript is repetitive, often vague in the terminology and does not seem to follow a systematic approach for the reader. Whilst there are some interesting points and findings it is hard to untangle the main aims and messages with this manuscript. It requires extensive editing for English, clarification of techniques, figures and tables, and expansion of adequate references to substantiate the authors statements.
Below are some issues/comments regarding this manuscript for the authors to address:
GENERAL COMMENTS:
Please extensively correct your English.
Please ensure you define all acronyms prior to referring to them.
Please ensure you reference your figures in your text and label your x and y axis on all relevant graphs.
INTRODUCTION: Your introduction is repetitive. Please refine to avoid duplication of content.
LINE 43: Your statement “As a result, any growth of such unusual cells can cause issues.” requires further clarification and references.
LINES 45-46: Your statement “there is indeed a chance that the pressure within the mind will lead to human death” require more a precise statement providing the probability/proportions.
LINE 54: Your statement “Better and healthier tissues from areas affected” requires to be more specific. Please clarify and expand
LINES 60-61: Please provide a reference for your statement “A usual brain tumor can double in size in just twenty-five days.”
LINES 61-62: Please provide valid references for each of your statements “If the person is not treated correctly, the 61 person's survival rate is typically less than 12 months. It can quickly lead to death.”
LINES 83-89: This paragraph is unclear and vague. For example- “DL is newer and more practical version of Machine learning and artificial intelligence.” Neural networks date back to the early 20th century. Please refine your introduction to deep learning.
LINES 100-101: Please provide a reference for your statement “A ConvNet takes considerably less pre-processing than other 100 classification techniques”
LINES 103-112, LINES 114-127: Please provide a diagram outlining the process described in these paragraphs.
LINES 120-121: Your sentence “The importance of having tiny kernel sizes is that overfitting mechanism.” Seems unfinished or does not make sense. Please correct.
LINES 121-122: Your sentence “In neural networks, optimization algorithms are being used to evaluate not whether a neuron must be engaged by determining the weighted sum of input.” Seems unfinished – please correct.
SECTION 1.1.3. – This section is very disjointed and confusing. Please revise to be more succinct and provide an example diagram to explain the method.
LINES 144-146: Your sentence “Because matching a brain with a big tumor area to an image is complex, some algorithms included atlas registration in contrast to tumor segmentation, Arabian Journal for Science and Engineering (2019)” seems to be using a reference in an unconventional format.
LITERATURE REVIEW: Your literature review is repetitive and mixes methods within it. This section requires extensive revision to be more relevant, clearer and succinct.
LINES 186-187: Your statement “This can lead to brain damage, which 186 is fatal.” Is not correct. Minor traumatic brain injury is not fatal.
LINE 192: You reference a “particular result” but it is unclear as to which result you refer to.
TABLE 1: This table is inconsistent with Findings/Results and does not specify some of the figures (e.g. “87% and 92%” for no 1).
LINE 320-322: You reference imbalance and that ‘accuracy won’t help much’ but there are a number of ML techniques to deal with imbalance. Please discuss the impact of these techniques.
SECTION 4.1: Please outline the Python packages use cite the authors.
LINE 354: Your sentence “Due to progressions generated by the matter….” Is ambiguous as it does not specify ‘matter’. Please correct the English.
SECTION 4.3: This is repetitive.
FIGURES 2 & 3: Please explain what A and B is in the figure description and reference this figure in your text above.
Formulas 11 and 12: This formula seems incomplete and incorrect in its current format. Please correct.
TABLE 2: You have bundled Precision and Accuracy together in the one column heading but they do not mean the same thing. Please clarify.
LINE 510: “Therefore, the table…” Please clarify which table you are referring to.
DISCUSSION: Please include a paragraph on the strengths and limitations of your work.
Author Response
Thank you so much for your valuable time in reviewing our manuscript. I am attaching a review report as per your valuable suggestions. Please let us know if you have any other valuable suggestions for our manuscript.
Thank you

Reviewer 2 Report
This manuscript does not in-depth of describing the procedure of data processing, e.g. how to input the original data to clustering and feature selection. Thus, it is not easy for other researchers to understand and reproduce the method.
The novelty and difference between this work and related works should be discussed.
The presentation of this work is very poor and would require heavy editing before being considered for publication. There are numerous sentences that hardly make any sense
Author Response

(The authors gave the same response as above.)

Reviewer 3 Report
Too much technical details are presented without much guidance of the reader through what is shown and why. This needs major improvement for better clarify of the presentation.
2. The most key contributions and significance should be highlighted and ordered with their importance.
3. Please, clearly explain how your solution advances existing approaches,
4. Some paragraphs are too long. Please divide them into several short paragraphs to improve the readability.
5. Acronyms must be defined when they are first used in the text
6. The author should introduce the proposed approach in more detail (in the abstract)
7. Every time a method/formula is used for something, it needs to be justified by either (a) prior work showing the superiority of this method, or (b) by your experiments showing its advantage over prior work methods - comparison is needed, or (c) formal proof of optimality. Please consider more prior works.
8. The authors should use more discussions to clearly explain how they designed the algorithm, and why / when the steps can play a role.
9. Tables where results are compared with other papers should contain reference numbers
1. Does the proposed method have some shortcomings? In fact, shortcomings don’t reduce the availability of the proposed method. By contrast, it is a very suitable way to help readers to understand the proposed method comprehensively in my opinion.
1. In the Conclusion section, please explain more about future works
Author Response

(The authors gave the same response as above.)

Round 2
Reviewer 2 Report
Now it is shaped and most of the section polished .
Still in abstract required to be more polish . It seems long passage .
Author Response
Thank you so much for your valuable time. Please check the attached file.
Thank you

This manuscript is a resubmission of an earlier submission. The following is a list of the peer review reports and author responses from that submission.